# Impact of COVID-19 Pandemic on Mental Health among the Population in Jordan

**DOI:** 10.3390/ijerph20146382

**Published:** 2023-07-17

**Authors:** Ghaith Salameh, Debbi Marais, Rawan Khoury

**Affiliations:** 1School of Health, University of Essex, Colchester CO4 3SQ, UK; 2Warwick Medical School, University of Warwick, Coventry CV4 7AL, UK; d.marais@warwick.ac.uk; 3Help Age International, London SE1 7RL, UK; khoury.rawan@gmail.com

**Keywords:** mental health, COVID-19 pandemic, Jordan, mental disorders, general population

## Abstract

Background: Mental health is a key indicator for public health measures. Jordan is one of the countries that has a high prevalence of mental illness and disorders. The COVID-19 pandemic has affected all health services in the country with a high refugee population. The aim of this review is to assess the impact of the pandemic on mental health in Jordan and identify key factors affecting it, in addition to addressing lessons learned from the pandemic. Methods: A systematic search was conducted on Medline Plus, Embase, Web of Science, EBSCOHost Psycinfo and Cinhal, following the PRISMA guidelines. Articles were selected based on predefined inclusion and exclusion criteria. Data were extracted and synthesized using narrative descriptive analysis. Results: The pandemic had a significant impact on PTSD, psychological distress, anxiety, depression and stress. Predictors of a higher impact on mental health were related to gender, socio-economic status and comorbidities. The healthcare workers group was the most affected by mental disorders. Conclusions: The COVID-19 pandemic’s impact on mental health was associated with high levels of PTSD, anxiety, depression and stress. In a country with a high prevalence of mental disorders, prompt and quick measures are needed to support the health system to absorb the effect of the pandemic and be responsive to dealing with the existing high prevalence.

## 1. Introduction

Mental health is a key indicator for public health measures in any country. The global burden of mental disorders and illnesses has been increasing proportionally and is estimated to have increased by 48.1% from 1990 to 2019 [1]. Health systems in all countries are facing challenges to address this mental disorder burden. Despite this, mental health services are not prioritized and are given the least support from policy makers [2].

Jordan has a high prevalence of mental disorders, around 26.1%, putting the country on the higher end of the world range, which is between 5.0% to 27.0% [3]. The country is still in the early phase of service provision for this. In 2010, the World Health Organization (WHO) selected Jordan as one of six countries to pilot the implementation of the mental health gap action program; a program that supports and enables countries to strengthen their mental health services [4].

It is also known that Jordan hosts around 3 million refugees from neighboring countries due to conflicts and wars. Around 600,000 refugees have entered Jordan since the beginning of the Syrian conflict in 2011, resulting in the second largest refugee camp in the world [5]. This has put a huge strain on the health system and elevated the burden of mental health disorders, as it is well recognized that such a burden increases within post-emergency response areas and their displaced population [6].

The COVID-19 pandemic reached the entire world by 2021, making it the worst health crisis that humans have faced in recent history, especially in terms of mental health [7]. Jordan took stringent measures to combat the pandemic, and the population faced total lockdown, social distancing, job loss, economic burden, insecurity and much more. Since there was already a high prevalence of mental disorders pre-pandemic in Jordan, it was expected that the post-pandemic impact would be high and have a multi-layer effect. This raised the question of how policy makers in Jordan would tackle the issue of mental health, and what support would be needed for the health system and for vulnerable communities to tackle this challenge.

Worldwide, various studies have reviewed the impact of the COVID-19 pandemic on the population level [8,9,10,11,12,13,14,15], and several have studied the impact on sub-population groups such as healthcare workers (HCWs) and refugees [16,17]. A scoping review of the mental health research during the pandemic in the Arab region [8] shows that there was a shortage of quality research on mental health in the region and it requires further attention to ensure better outcomes on the mental well-being of the population.

Alzahrani and colleagues [9] reported that in the neighboring country of Saudi Arabia, the overall prevalence of depression, anxiety and stress was 30%, 20% and 29%, respectively. A higher prevalence was associated with sociodemographic risk factors, such as being female, having a younger age or unemployment. Other risk factors were health-related such as previous psychiatric conditions or fear of COVID-19. A high prevalence rate of stress, between 8.1% and 81.9%, has also been reported during the pandemic globally [10].

A meta-analysis evaluated the prevalence of mental health issues among HCWs [16]. Results from 45 papers showed that PTSD was the most prevalent at 21.7%, followed by anxiety disorder at 16.1%. This review included studies from a wide range of countries globally and various health disciplines.

On the refugee population level, which is a population segment that is understudied and about which little is presented, a qualitative systematic narrative study [17], reviewed 16 studies concerned with refugees and the impact of the pandemic on mental health symptoms. Being displaced due to war, conflicts or other negative life-changing events put refugees among the highest affected groups of PTSD and other mental disorders. With the pandemic, it was even worse for this vulnerable population group. The most affected were the younger group.

Thus, the aim of this review was to identify the impact of the COVID-19 pandemic on the mental health of the population in Jordan, which is unknown at this stage. The impact was reviewed on all population segments, identifying the prevalence of mental disorders among the Jordanian population. A further objective was to make recommendations on how mental health services, policies and response can be enhanced and what lessons can be learned from the pandemic’s effect on mental health.

## 2. Materials and Methods

Methods and results were formatted based on the PRISMA reporting guidelines [18]. The PRISMA 2020 main checklist was included as a Appendix A. The protocol of this systematic review was registered on PROSPERO under the number CRD42022331561.

### 2.1. Search Strategy

A structured literature search was conducted using the following databases: Medline Plus, Embase, Web of Science, EBSCOHost Psycinfo and Cinhal. The Cochrane database of systematic reviews was checked to confirm that no existing systematic review had been conducted on the same topic.

The following search terms were used (Jordan OR Hashemite Kingdom of Jordan OR Jordanian) AND (Mental Health OR Mental Disorders OR Mental illness OR Psychiatric illness OR Depression OR Anxiety) AND (COVID-19 OR Coronavirus OR 2019 nCov OR SARS-CoV-2 OR COVID OR Pandemic) AND (General population OR general public OR Public OR community), including Boolean operators as indicated.

### 2.2. Study Selection and Eligibility Criteria

Inclusion criteria were as follows: (1) the population level included all groups such as, but not limited, to refugees, vulnerable communities, females, university students and HCWs. This inclusion ensured a comprehensive outcome of the effect of the pandemic on the whole population. (2) All types of study designs were included. (3) The review included all articles since the beginning of the pandemic, i.e., from March 2020 to May 2022, the date of the final search.

This research excluded the following: (1) any article that did not include only Jordan or a specific sample for Jordan; (2) literature review articles; (3) Arabic language articles. This bias of language choice did not affect the final research outcomes since health research publishing and education in Jordan is in the English language [19].

### 2.3. Screening

Rayyan.ai software was used for screening as it is considered an application of significance and supports the process of data management during the systematic review process [20]. During the first phase of screening, duplicates were removed. The remaining articles were then screened on the basis of the title and abstract. The full-text articles of the remaining records were then uploaded on Rayyan. A final screening for the remaining full-text articles was conducted rigorously to ensure compliance with the review key elements, inclusion and exclusion criteria.

Both phases of screening were conducted by two independent reviewers (GS and RK), and a third reviewer (DM) resolved any conflicting decisions to ensure proper selection of eligible studies and non-selection bias [21].

### 2.4. Data Extraction

A data extraction form was created and included the following data: (1) title, (2) lead author and year of publication, (3) population group, (4) study design, (5) sample size, (6) sample characteristics, (7) assessment tools, (8) outcomes and (9) prevalence of symptoms of depression/anxiety/post-traumatic stress disorder (PTSD)/psychological distress/stress.

Key elements of the data extracted included the following: (1) the population group to be able to categorise the different studies per the population groups in Jordan, (2) the assessment tools used to determine the prevalence of mental disorders and (3) the prevalence of symptoms of depression/anxiety/ PTSD/psychological distress/stress.

### 2.5. Quality Appraisal

The Critical Appraisal Skill Program (CASP) tool was chosen for randomized clinical trials (RCT). For cross-sectional (CS) studies, the AXIS tool was used. A final analysis was created, and the scores were recorded. All studies on mental health in Jordan eligible in this review were included, with no exclusion after appraisal.

### 2.6. Data Synthesis

The narrative descriptive analysis method was chosen for data analysis. In order to exclude any bias, a well-informed and structured process of analysis was performed using the guidelines from the Cochrane consumers and communication review group on data synthesis and analysis [22].

## 3. Results

### 3.1. Search Results

In total, 260 publications were identified from searching the databases (Figure 1). Out of the 260, n = 106 publications were initially removed due to duplication. The remaining n = 154 publications were screened on a title and abstract basis. Ninety-five publications were excluded after the title and abstract screening. The agreement rate in the title and abstract phase screening between the first and second reviewer was 86%. Then, the third reviewer (DM) resolved the conflict, resulting in a final agreement of 59 records. The remaining articles were assessed for eligibility based on full-text screening.

Out of the 59 publications, n = 26 was excluded according to the exclusion criteria. Publications removed were as follows: n = 15 were excluded for studying different outcomes and results. Nine publications (n = 9) were excluded for having a different population group than Jordan. Two publications (n = 2) were removed due to non-retrieval. One publication (n = 1) was removed for being a review paper. One publication (n = 1) was removed for being a commentary paper. The overall agreement rate for the full-text screening phase between the first and second reviewer was 80% before the final conflicts were resolved by the third reviewer, resulting in 95% agreement.

### 3.2. Study Characteristics

Table 1 summarises the extracted study characteristics. The sample size of the 31 studies ranged from 26 to 6157 participants, with a total of 42,357 participants. A variety of population groups was studied, including (1) refugees and vulnerable communities (one RCT study and the other CS) with a total of 3721 participants; (2) university students with seven CS studies and a total of 12,750 participants, and one CS study with 382 university teachers; (3) HCWs with five CS studies and a total of 3104 participants and (4) older adults 60+ with three CS studies and a total of 1019 participants. The remaining studies were non-specific and included the general population.

The sample characteristics between studies varied; the main characteristics extracted were the age and male-to-female (m/f) ratios. For the refugees and vulnerable communities, one RCT study had a mean age of 40.4, and the other study targeted adolescents living in camps or tented settlements within hosting communities aged 10–17 years. For the university student population subgroup, age ranges were between 18 and 24, with a higher female ratio mostly across all studies in this subgroup. For the HCWs subgroup, the mean age varied but mostly represented younger age groups, with an m/f ratio that varied across the studies. For the older adults subgroup, the first study had an age range between 60 and 68, the second a mean age of 72.4 and the third a mean age of 67.6; m/f ratios were mostly comparable.

The rest of the studies that researched the general population have a variation of the mean ages but mostly the mean age falls within the younger representation, while the m/f ratio is comparable in general.

The outcomes varied among the different studies. Most of the studies included measures of anxiety, depression and stress, with different measuring tools. These amounted to a total of 22 studies. Four studies measured symptoms of PTSD, while two studies assessed psychological distress. It was observed that three studies did not include direct measures of the prevalence of mental health disorders: one study measured death anxiety/spiritual well-being/religion coping in older adults related to COVID-19; another study looked for gender-based disparities by checking psychiatric illness and how gender is a predictor of depression; the final one evaluated the Arabic version of the Fear of COVID-19 Scale (FCV-19S)

### 3.3. Quality Appraisal

The results of the quality appraisal for all the CS studies using the AXIS tool are presented in Table 2. The overall quality of the included studies was moderate, with scores varying from twelve to twenty points (out of a possible twenty points). There were two studies with a red flag since there was no information on funding or ethical approval. Two additional studies had a low score, one with twelve; this score was due to a lack of information about the targeted sampling reference and the lack of sufficient information on the statistical methods used. The other low-scoring study of thirteen was due to a lack of consistency of results and the limitations of the study being presented.

The moderate scoring range is between fourteen and seventeen for a total of eighteen studies. Most of the studies within this range have basic research requirements and information. The lack of proper justification of sample size, measure of addressing non-responders and the response rates related to it, were the reasons for a moderate scoring.

The scoring for the remaining studies was as follows: four studies with a score of eighteen and three studies with a score of nineteen, and this was due to further information on the non-responders and response rates. One study scored twenty, but this was exceptional since it had all the elements covered within the AXIS critical appraisal tool. The study was funded by the research and evaluation division of the UK Foreign Commonwealth and Development Office for the Gender and Adolescence: Global Evidence (GAGE) longitudinal study [23]. The study was part of the response to the Syrian refugee crisis in Jordan to evaluate the situation and build a more resilient program.

The one RCT study appraised using the CASP RCT tool for critical appraisal showed moderate quality. The study participants were not blinded, changes from the baseline group were also noticed, and *p*-values were reported but the confidence intervals (CI) were not. In general, the study provided vital information and results related to mental health within a very high-risk population setting.

### 3.4. Assessment Tools

There have been a variety of tools used to measure the prevalence of symptoms of depression/anxiety/PTSD/psychological distress/stress. It was also noticed that there were some additional tools used to measure other related outcomes. Table 3 lists all assessment tools used and shows the results related to the prevalence of symptoms associated with these tools.

The most used assessment tools were the Generalised Anxiety Disorder (GAD-7); it is a reliable measurement tool with a seven-item anxiety scale used for clinical and research needs [24]. This scale was used in six studies. The next most used assessment tool was the 10-item Kessler Psychological Distress Scale (K10), this scale measures the level of anxiety and stress symptoms for participants, it is a self-reported scale and is widely used for mental health assessments [25]. The (K10) scale was used in five studies and one study used the (K6) scale. The Depression, Anxiety and Stress Scale, a 21-item questionnaire (DASS-21), is a set of three self-report scales designed to measure the emotional states of depression, anxiety and stress. It was used in four studies, and in one study it was translated into Arabic. The Fear of COVID-19 Scale (FCV-19S) was developed after the COVID-19 pandemic and has been widely used to assess the fear of COVID-19 infections and anxiety. Its psychometric properties are well studied [26], and it was used in four studies. Finally, the nine-item Patient Health Questionnaire (PHQ-9) assesses depression and its severity [27]. This scale was used in three studies.

A variety of other prevalence of anxiety, stress and PTSD assessment tools were less frequently used among the different studies. Figure 2 illustrates the tools used and their frequency in total.

Apart from the prevalence of symptoms of depression/anxiety/PTSD/psychological distress/stress assessment tools used in the different studies, analysis showed that there were other instruments used for assessments to support the different research outcomes. These tools included disability measurement, coping with stress, sleep quality and food insecurity in addition to health and well-being. All of these are related outcome factors to issues that escalated during the COVID-19 pandemic lockdown and post lockdown.

### 3.5. Prevalence of Symptoms

#### 3.5.1. PTSD

The prevalence of PTSD was assessed in four out of the thirty-one studies [28,29,30,31]. The first study on the refugee subgroup showed that there was a greater decrease in PTSD severity in people assessed during the pandemic than in those assessed prior to the pandemic. The second study included nurses who worked with COVID-19 patients, and it showed that the prevalence of PTSD was 37.1%, while the majority were at the lowest level of PTSD at 17%. The third study highlighted that 3.1% of the general population was diagnosed with PTSD. The final one was on older adults and indicated that COVID-19 had a moderate impact on symptoms of PTSD, and older adults with comorbidities had higher levels of PTSD symptoms.

#### 3.5.2. Psychological Distress

Two out of the thirty-one studies assessed psychological distress [32,33]. The first study was a CS study across 17 countries among the general population. Jordan demonstrated statistically significant high psychological distress compared to the baseline country, which was Thailand in this study. Concerning the (K10) psychological distress, the low distress ratio was 14.9%, while the moderate to high ratio was 85.1%. The second study assessed the psychological distress among university students. Concerning the (K10) psychological distress scale prevalence, 69.5% were severe, 12.6% moderate, 10.8% mild and 7.1% none. Which also showed significant psychological distress.

#### 3.5.3. Anxiety, Depression and Stress

Twenty-two studies out of the thirty-one studies assessed elements of mental health related to anxiety, stress and depression. Most of the studies showed increased levels of anxiety, depression and stress within the different population groups during the COVID-19 pandemic.

Five studies assessed the three elements of anxiety, depression and stress [34,35,36,37,38]. The first study reported the prevalence of depression, anxiety and stress at different levels of 78.7%, 67.9% and 58.7%, respectively. Related predictors of higher levels were associated with home quarantine. A correlation was found between higher levels of depression, anxiety and stress and demographic, health-related factors and lifestyle variables. The second study reported that 49.2% of participants experienced increased anxiety, 72.4% experienced increased worry, 23.1% experienced increased depression and 22.6% experienced increased panic. Females were shown to be more susceptible to emotional and mental stress. In the third study, participants reported 57.5% depression, 42.0%, stress and 59.1% anxiety. This study indicated that risk factors related to increased distress scores were female gender, married people, knowing someone who died from COVID-19 and worrying about COVID-19. The fourth study reported a prevalence of depression of 57.8%, anxiety of 42.4% and stress of 50.1%. Statistically significant risk factors were related to gender, number of dependents and caring for a COVID-19 patient. The fifth study reported symptoms of depression, anxiety and stress at 41.8%, 24.5% and 22.8% to be mild, extreme and severe, respectively, with the increasing levels of symptoms related to the fear of COVID-19 score.

Five studies assessed the prevalence of anxiety and depression [30,39,40,41,42]. In the first study and in terms of anxiety, 53.0% of the participants reported symptoms: 33.8% mild, 12.9% moderate and 6.3% severe anxiety. Half of the respondents reported depressive symptoms. All these symptoms were related to poor sleep quality among participants. In the second study, 8.9% of participants were diagnosed with anxiety disorder, and 5.6% were diagnosed with depression. Predictors of mental disorders were low monthly income, unemployment and diabetic patients. The third study reported a depression prevalence of 23.8% and an anxiety prevalence of 13.1%. Higher prevalence was noticed among the risk groups of females, divorced people and those having chronic illnesses. The fourth study reported the prevalence of anxiety symptoms at 33.8% mild, 12.9% moderate and 6.3% severe. The prevalence of depression symptoms was as follows: 21.5% highest quartile, 26.8% third quartile, 24.8% second quartile and 26.9% lowest quartile. The fifth exploratory study reported higher level of anxiety and depression among HCW’s with high workload and in isolation units. 

Two studies assessed stress and depression prevalence [23,43]. The first study, on the refugee population, reported that 19.3% of adolescents presented with symptoms of moderate-to-severe depression, and two-thirds of adolescents reported that household stress had increased during the pandemic. The second study reported an overall increased stress level and an increased feeling of fear and anxiety. This was a phenological method study.

The remaining 11 studies, out of the 22, assessed one element of either anxiety, stress or depression [44,45,46,47,48,49,50,51,52,53,54]. As shown in Table 3, all the studies reported a moderate, high or severe prevalence of anxiety, depression and stress associated with COVID-19. Higher levels of prevalence were associated with gender—being female—or being an older member of a population.

#### 3.5.4. Non-Prevalence

Three studies did not include direct measures or prevalence of mental health [55,56,57]. The first study measured coping with death anxiety/spiritual well-being/religion in older adults in the context of COVID-19. The second study looked for gender-based disparities by checking psychiatric illness and concluded that gender is a predictor of depression. The final study evaluated the Arabic version of the Fear of COVID-19 Scale for use in other research related to mental health and COVID-19 and concluded that the Arabic FCV-19S is a reliable scale.

**Table 1 ijerph-20-06382-t001:** Data extraction summary.

No.	Title	Lead Author and Year of Publication	Population Group	Study Design	Sample Size	Sample Characteristics	Outcomes
1	A longitudinal study of mental health before and during the COVID-19 pandemic in Syrian refugee	(Akhtar et al., 2021) [28]	Refugees	RCT	410	Mean Age (40.4), m/f Ratio 28.9%/71.1%	Primary: total score for anxiety and depression measured by (HSCL-25). Secondary: PTSD symptoms assessed by (PCL-5)
2	Anxiety and coping strategies among nursing students returning to university during the COVID-19 pandemic	(Masha’al et al., 2022) [44]	University nursing students	CS	282	Nursing university students, m/f ratio: 25.9%/74.1%	Higher anxiety levels were reported by female nursing students and students who had fear of contracting the virus. Using coping strategies outcomes showed that anxiety correlated positively with denial, behavioural disengagement, venting and self-blame
3	Anxiety and depressive symptoms are associated with poor sleep health during a period of COVID-19-induced nationwide lockdown: a cross-sectional analysis of adults in Jordan	(Al-Ajlouni et al., 2020) [39]	Adults	CS	2202	Age 18–65, mean age (37.35), m/f ratio: 52.9%/ 47.1%	Increased level of anxiety and depressive symptoms and their association with poor sleep health outcomes
4	Association of Death Anxiety with Spiritual Well-Being and Religious Coping in Older Adults During the COVID-19 Pandemic	(Rababa et al., 2021) [55]	Older Adults (60–75)	CS	248	Age ranges: 60–68, m/f ratio: 42.3%/57.7%	Death Anxiety, Spiritual well-being, religious coping, secondary purpose examining the differences in main variables based on sociodemographic characteristics
5	Attitudes, Anxiety, and Behavioural Practices Regarding COVID-19 among University Students in Jordan: A Cross-Sectional Study	(Olaimat et al., 2020) [45]	University students	CS	2083	(62.6%) within the ages of 20–24.9 years, m/f ratio: 24.5%/75.5%	81.1% of students displayed positive attitudes toward COVID-19, 84.3% of students showed low-risk practices toward COVID-19, 69.2% of students were found to be anxious about being infected with the virus
6	Compounding inequalities: Adolescent psychosocial wellbeing and resilience in Jordan during COVID-19	(Jones et al., 2022) [23]	Refugees and vulnerable Jordanians	CS	(3311) total out of which (2574) for the panel sample	10–12 younger cohort, 15–17 older cohort, Adolescents living in camps or tented settlement within hosting communities	Pre-COVID-19 vulnerabilities, Post-COVID-19 disrupted social contexts, COVID-19 and adolescent psychosocial wellbeing, coping and resilience under COVID-19
7	COVID-19: Factors associated with psychological distress, fear, and coping strategies among community members across 17 countries	(Rahman et al., 2021) [32]	general community members,	CS	538 from Jordan (6.3% of sample size)	Adults aged ≥ 18 years, mean age of (33.3), m/f ratio: 35.6%/64.4%	Psychological distress, levels of fear to COVID 19, coping strategies
8	COVID-19–Related Posttraumatic Stress Disorder Among Jordanian Nurses During the Pandemic	(Qutishat et al., 2021) [29]	HCWs	CS	259	Age range: 23 to 58 y, more than half aged between 25 and 34 y old (53.3%; n = 138), m/f ratio: 52.1%/47.9%	PTSD related to nurses working with patients diagnosed with COVID-19
9	Depression and coping among COVID-19-infected individuals after 10 days of mandatory in-hospital quarantine, Irbid, Jordan	(Samrah et al., 2022) [46]	Adults	CS	66	Older than 18 years, mean age (35.8), m/f ratio: 40.9%/59.1%	Depression, coping methods
10	Depression, anxiety and stress among undergraduate students during COVID-19 outbreak and “home-quarantine”	(Hamaideh et al., 2021) [34]	University Students	CS	1380	Mean age (20.8), m/f ratio: 23.9%/76.1%	Prevalence of depression, anxiety and stress, and the predictors related to demographics
11	Depression, coping skills, and quality of life among Jordanian adults during the initial outbreak of COVID-19 pandemic: cross-sectional study	(Al-Shannaq et al., 2021) [38]	Adults	CS	511	Age: 18–65, mean age: (30), m/f ratio: 34.8%/ 65.2%	Establishing the link between psychological issues and the COVID-19
12	Effect of COVID-19 Quarantine on the Sleep Quality and the Depressive Symptom Levels of University Students in Jordan During the Spring of 2020	(Saadeh et al., 2021) [48]	University students	CS	6157	Mean age (19.79), m/f ratio:28.7%/71.3%	Sleep Quality, Depressive symptoms
13	Evaluating the impact of COVID 19 on mental health of the public in Jordan: A cross-sectional study	(Suleiman et al., 2022) [30]	Adults	CS	1820	63.5% aged 30–55, m/f ratio: 44.2%/55.8%	Psychiatric Disorders, Anxiety, psychological stress, predictors of mental preparedness for pandemic
14	Gender-based disparities on health indices during COVID-19 crisis: a nationwide cross-sectional study in Jordan.	(Abufaraj et al., 2021) [56]	Adults	CS	1300	Mean age: (43), m/f ratio: 50.5%/49.5%	Disparities between genders in health indices, mental well-being and economic burden
15	Is It Just About Physical Health? An Online Cross-Sectional Study Exploring the Psychological Distress Among University Students in Jordan in the Midst of COVID-19 Pandemic	(Al-Tammem et al., 2020) [33]	University Students	CS	381	Mean age (22.8), m/f ratio: 47.8%/52.2%	Psychological distress, motivation towards distance learning, coping activities
16	Loneliness and Depression among Community Older Adults during the COVID-19 Pandemic: A cross-sectional study	(Alhalaseh et al., 2022) [49]	Older adults (60 y and older)	CS	456	Mean age: (72.48), m/f ratio: 49.8%/50.2%	Development of Loneliness and depression, factors affecting those outcomes in the older adults’ communities
17	Medical students’ relative immunity, or lack thereof, against COVID-19 emotional distress and psychological challenges; a descriptive study from Jordan	(Kheirallah et al.,2021) [35]	Medical students	CS	1404	m/f ratio: 40.1%/59.9%	Changes in emotional reactions due to COVID-19, effect of social media usage on the emotional distress during COVID-19
18	Mental health impacts of COVID-19 on healthcare workers in the Eastern Mediterranean Region: a multi-country study	(Ghaleb et al., 2021) [36]	HCWs	CS	55 from Jordan (3.8%) from total sample	52.7% less than 30 yrs., m/f ratio: 51.2%/48.8%	Prevalence of depression, anxiety and stress among HCWs responding to COVID and related associated factor
19	Mental health status of the general population, healthcare professionals, and university students during 2019 coronavirus disease outbreak in Jordan: A cross-sectional study	(Naser et al., 2020) [40]	General population,	CS	4126	55.4% aged 18 to 29, m/f ratio: 41%/59%	Prevalence of depression and anxiety among GP’s, HCW and university students, identify key population who need psychological intervention
20	Prevalence and predictors of depression, anxiety, and stress among Jordanian nurses during the coronavirus disease 2019 pandemic	(Al-Amer et al., 2021) [37]	Nurses	CS	405	Mean age: (30.27), m/f ratio: 28.6%/71.4%	Prevalence of anxiety, depression and stress
21	Prevalence Estimates and Risk Factors of Anxiety among Healthcare Workers in Jordan over One Year of the COVID-19 Pandemic: A Cross-Sectional Study	(Yassin et al., 2022) [50]	HCWs	CS	422	Mean age: (35.5), m/f ratio: 71.3%/28.7%	Prevalence estimates, severity, and risk factors of anxiety among healthcare workers
22	Psychological Impact of COVID-19 Pandemic Among the General Population in Jordan.	(Khatatbeh et al., 2021) [43]	General population	CS	2854	m/f ratio: 41.4%/58.6%	Assessment of COVID impact on population, evaluation of sociodemographic influence on the impact
23	Psychological impacts during the COVID-19 outbreak among adult population in Jordan: A cross-sectional study	(Al-Shannaq et al., 2021) [47]	Adults	CS	725	Mean age: (33.7), m/f ratio: 43.6%/56.4%	Change in Daily life experience during COVID, Psychological impact of COVID (fear of COVID, anxiety, stress, depression), Gender based differences in psychological impacts, age related correlation of psychological impacts, factors predicting fear of COVID
24	Psychometric Properties of the Arabic Version of the Fear of COVID-19 Scale (FCV-19S) Among Jordanian Adults	(Al-Shannaq et al., 2021) [57]	Adults	CS	725	Mean age: (33.7), m/f ratio: 43.6%/56.4%	Assessment of the Arabic version for the (FCV-19S) and its validation to report fear of COVID among adults
25	Staying Physically Active Is Associated with Better Mental Health and Sleep Health Outcomes during the Initial Period of COVID-19 Induced Nation-Wide Lockdown in Jordan	(Al-Ajlouni et al., 2022) [41]	Adults	CS	1240	Mean age: (37.4), m/f ratio: 52.9%/47.1%	Prevalence of physical activity and its relation to mental health and sleeping among Jordanian
26	The Experiences of Nurses and Physicians Caring for COVID-19 Patients: Findings from an Exploratory Phenomenological Study in a High Case-Load Country	(Khatatbeh et al., 2021) [42]	HCWs	CS	26	Mean age: (29.9), m/f ratio: 61.5%/38.5%	HCWs in Jordan working in the wards and care centres designated for patients with COVID-19 experienced mental and emotional distress
27	The impact of confinement on older Jordanian adults’ mental distress during the COVID-19 pandemic: A web-based cross-sectional study	(Abu Kamel et al., 2021) [31]	Older adults 60+	CS	315	Mean age: (67.6), m/f ratio: 58.7%/41.3%	Psychological impacts of confinement, factors affecting PTSD
28	The impact of the COVID-19 pandemic on mental health: early quarantine-related anxiety and its correlates among Jordanians	(Massad at al., 2020) [51]	Jordanian 18+	CS	5274	Highest age group is 18–24, m/f ratio: 44.7%/55.3%	Prevalence of quarantine related psychological distress, and its correlation with sociodemographic
29	The Impact of the COVID-19 Pandemic and Emergency Distance Teaching on the Psychological Status of University Teachers: A Cross-Sectional Study in Jordan	(Akour et al., 2020) [52]	University Teachers	CS	382	Mean age: (43.9), m/f ratio: 55.5%/44.5%	Assessment psychological distress, Challenges related to online learning and their psychological impact, Self-coping activities among teachers
30	The inevitability of COVID-19 related distress among healthcare workers: Findings from a low caseload country under lockdown	(Hawari et al., 2021) [53]	HCWs	CS	937	Mean age: (33.3), m/f ratio: 43.9%/56.1%	Distress among HCW in a low caseload country
31	The Prevalence of Mental Distress and Social Support among University Students in Jordan: A Cross-Sectional Study	(Abuhamdah et al., 2021) [54]	University Students	CS	1063	Age between 18–24: 71%, m/f ratio: 29.2%/70.8%	Prevalence of mental distress, perceived social support from university student’s perspective

**Table 2 ijerph-20-06382-t002:** Results from the quality appraisal (AXIS) tool.

Study No.	(Masha’al et al., 2022) [44]	(Al-Ajlouni et al., 2020) [39]	(Rababa et al., 2021) [55]	(Olaimat et al., 2020) [45]	(Jones et al., 2022) [23]	(Rahman et al., 2021) [32]	(Qutishat et al., 2021) [29]	(Samrah et al., 2022) [46]	(Hamaideh et al., 2021) [34]	(Al-Shannaq et al., 2021) [38]	(Saadeh et al., 2021) [48]	(Suleiman et al., 2022) [30]	(Abufaraj et al., 2021) [56]	(Al-Tammem et al., 2020) [33]	(Alhalaseh et al., 2022) [49]	(Kheirallah et al.,2021) [35]	(Ghaleb et al., 2021) [36]	(Naser et al., 2020) [40]	(Al-Amer et al., 2021) [37]	(Yassin et al., 2022) [50]	(Khatatbeh et al., 2021) [43]	(Al-Shannaq et al., 2021) [47]	(Al-Shannaq et al., 2021) [57]	(Al-Ajlouni et al., 2022) [41]	(Khatatbeh et al., 2021) [42]	(Abu Kamel et al., 2021) [31]	(Massad at al., 2020) [51]	(Akour et al., 2020) [52]	(Hawari et al., 2021) [53]	(Abuhamdah et al., 2021) [54]
**Introduction**																														
Clear aims and Objectives	Y	Y	Y	Y	Y	Y	Y	Y	Y	Y	Y	Y	Y	Y	Y	Y	Y	Y	Y	Y	Y	Y	Y	Y	Y	Y	Y	Y	Y	Y
**Methods**																														
Appropriate study design	Y	Y	Y	Y	Y	Y	Y	Y	Y	Y	Y	Y	Y	Y	Y	Y	Y	Y	Y	Y	Y	Y	Y	Y	Y	Y	Y	Y	Y	Y
Justified sample size	N	N	Y	Y	Y	Y	Y	N	Y	N	N	Y	Y	N	N	Y	N	Y	Y	N	N	Y	N	N	N	Y	N	Y	N	Y
Target population defined	Y	Y	Y	Y	Y	Y	Y	Y	Y	Y	Y	Y	Y	Y	Y	Y	Y	Y	Y	Y	Y	Y	Y	Y	Y	Y	Y	Y	Y	Y
Sample frame represent the reference population	Y	N	N	Y	Y	Y	N	N	Y	Y	Y	Y	Y	N	Y	Y	Y	Y	?	Y	Y	Y	Y	Y	Y	Y	N	Y	Y	Y
Selection process likely to select participants representing the reference population	Y	N	Y	Y	Y	Y	N	N	Y	Y	N	Y	Y	N	Y	?	Y	Y	?	Y	N	Y	Y	Y	Y	Y	N	Y	N	Y
Measures addressing non-responders	N	N	Y	N	Y	N	N	N	N	N	?	N	N	N	N	Y	N	?	N	Y	N	Y	N	N	N	Y	N	N	N	N
Measuring risk factor and outcome variable	Y	Y	Y	Y	Y	Y	Y	Y	Y	Y	Y	Y	Y	Y	Y	N	Y	Y	N	Y	Y	Y	Y	Y	Y	Y	Y	Y	Y	Y
Using measurement tools for risk factor and outcome variables	Y	Y	Y	Y	Y	Y	Y	Y	Y	Y	Y	Y	Y	Y	Y	N	Y	Y	Y	Y	Y	Y	Y	Y	?	Y	Y	Y	?	Y
Tools for statistical significance	Y	Y	Y	Y	Y	Y	Y	Y	Y	Y	Y	Y	Y	Y	Y	Y	Y	Y	Y	Y	Y	Y	Y	Y	?	Y	Y	Y	Y	Y
Methods (including statistical methods) sufficiently described	Y	Y	Y	Y	Y	Y	N	Y	Y	Y	Y	Y	Y	Y	Y	N	Y	Y	N	Y	Y	Y	Y	Y	Y	Y	Y	Y	Y	Y
**Results**																														
Adequate description of basic data	Y	Y	Y	Y	Y	Y	Y	Y	Y	Y	Y	Y	Y	Y	Y	Y	Y	Y	Y	Y	Y	Y	Y	Y	Y	Y	Y	Y	Y	Y
Response rate raise and non-response bias?	N	Y	N	N	N	N	?	Y	Y	?	N	?	N	?	N	N	?	?	?	N	N	N	N	Y	N	Y	?	N	?	N
Information about non-responders	N	N	N	N	Y	N	N	Y	N	N	N	N	N	N	N	N	N	N	N	N	N	N	?	N	?	N	N	Y	N	N
Consistent results	?	Y	Y	Y	Y	Y	Y	Y	Y	Y	Y	Y	Y	Y	Y	N	Y	Y	Y	Y	Y	Y	Y	Y	?	Y	Y	Y	Y	Y
Results presented for all the analyses	Y	Y	Y	Y	Y	Y	Y	Y	Y	Y	Y	Y	Y	Y	Y	Y	Y	Y	N	Y	Y	Y	Y	Y	Y	Y	Y	Y	Y	Y
**Discussion**																														
Justified discussions and conclusions	Y	Y	Y	Y	Y	Y	Y	Y	Y	Y	Y	Y	Y	Y	Y	Y	Y	Y	Y	Y	Y	Y	Y	Y	Y	Y	Y	Y	Y	Y
Study limitations discussed	Y	Y	Y	N	Y	Y	Y	Y	Y	Y	Y	Y	Y	Y	Y	?	Y	Y	Y	Y	Y	Y	Y	Y	Y	Y	Y	Y	Y	Y
**Other**																														
Funding sources, conflicts of interest	N	N	N	N	N	N	N	N	Y	N	?	N	N	N	N	N	N	N	N	N	N	N	N	N	N	N	N	N	N	N
Ethical approval or consent of participants	Y	Y	Y	Y	Y	Y	Y	Y	Y	Y	Y	Y	Y	Y	Y	Y	Y	Y	Y	Y	Y	Y	Y	Y	Y	Y	Y	Y	Y	Y
**Points**	15	14	16	17	20	18	14	16	16	16	15	17	18	14	17	13	16	17	12	18	16	19	17	16	14	19	14	19	14	18

**Table 3 ijerph-20-06382-t003:** Summary of assessment tools and prevalence.

No.	Lead Author and Year of publication	Assessment Tools	Prevalence of Symptoms of Depression/Anxiety/ PTSD/Psychological Distress/Stress
1	(Akhtar et al., 2021) [28]	(K10) (WHODAS) (HSCL-25) (PCL-5)	There was a greater decrease in PTSD severity in people assessed during the pandemic than those assessed prior to the pandemic
2	(Masha’al et al., 2022) [44]	(GAD-7) (Brief-COPE)	70.6% reported mild to severe anxiety levels upon returning to on-campus learning.
3	(Al-Ajlouni et al., 2020) [39]	(GAD-7) (CES-D) (PSQI) (IPAQ)	In terms of anxiety, 53% of the participants reported symptoms for mild (33.8%), moderate (12.9%) or severe anxiety (6.3%). Half of respondents reported depressive symptoms.
4	(Rababa et al., 2021) [55]	(ASDA) (SWBS) (BARCS)	N/A
5	(Olaimat et al., 2020) [45]	Questionnaire designed and developed based on the data available on the websites of the WHO, the CDC, and the European CDC (ECDC)	(69.2%) of the students were found to be anxious about being infected with the virus
6	(Jones et al., 2022) [23]	(GAGE) conceptual framework (PHQ-8) (GAD-7) (BRCS) (HFIAS)	19.3% of adolescents presented with symptoms of moderate-to severe depression, two thirds of adolescents reported household stress had increased during the pandemic
7	(Rahman et al., 2021) [32]	(K-10) (FCV-19S) (BRCS)	Jordan demonstrated statistically significant high psychological distress compared to the baseline country, (K10) low, moderate/ high ratio: 14.9%/85.1%
8	(Qutishat et al., 2021) [29]	(DSM-5) (PCL-5)	The prevalence of PTSD is (37.1%), The majority were at the lowest level of PTSD (17%), different prevalence rates related to different subgroups
9	(Samrah et al., 2022) [46]	(PHQ-9)	44% reported symptoms of depression, 21% are at high risk of major depressive disorder
10	(Hamaideh et al., 2021) [34]	Arabic version of (DASS-21)	Moderate/ high scores: depression (21.1/ 21.2), anxiety (24.3/16.8), stress (15.5/19.6)
11	(Al-Shannaq et al., 2021) [38]	(BDI-II) (COPE Scale) (WHOQOL-BREF)	35% minimal depression, 33% mild depression, 19% moderate depression, 13% severe depression
12	(Saadeh et al., 2021) [48]	(PSQI) (CES-D)	Prevalence of depressive symptoms 71% (34% for moderate and 37% for high depressive symptoms), 62.5% reported quarantine had a negative effect on their mental health
13	(Suleiman et al., 2022) [30]	(K10)	8.9% diagnosed with anxiety disorder, 5.6% diagnosed with depression, 3.1% diagnosed with PTSD
14	(Abufaraj et al., 2021) [56]	CSS local tool (PHQ-4) (UCLA) (TB) stigma scale	One-fourth had chronic medical or psychiatric illnesses, Gender is a significant predictor of higher PHQ-4 scores (women vs. men: β: 0.88, 95% CI: 0.54–1.22)
15	(Al-Tammem et al., 2020) [33]	(K10)	Psychological distress prevalence was: 69.5% severe, 12.6% moderate, 10.8% mild, 7.1% none
16	(Alhalaseh et al., 2022) [49]	(UCLA) (GDS)	Prevalence of loneliness post COVID was 41.1% compared to 14% pre-pandemic, depression prevalence: mild 23%, moderate: 7.7%, severe: 6.4%
17	(Kheirallah et aL.,2021) [35]	Self-reported questionnaire	49.2% experienced increased anxiety, 72.4% experienced increased worry, 23.1% with increased depression, 22.6% with increased panic
18	(Ghaleb et al., 2021) [36]	(DASS-21) (CPDI)	57.5% with depression, 42.0% with stress, and 59.1% with anxiety
19	(Naser et al., 2020) [40]	(PHQ-9) (GAD-7)	Prevalence of depression 23.8%, prevalence of anxiety 13.1%
20	(Al-Amer et al., 2021) [37]	(DASS)	Prevalence of Depression 57.8%, Anxiety 42.4%, Stress 50.1%
21	(Yassin et al., 2022) [50]	(GAD-7)	Prevalence of anxiety: mild (45%), moderate: (13.7%), severe (10%),
22	(Khatatbeh et al., 2021) [43]	(IES-R)	IES-R scores: 56.9% normal, 23.3% mild, 9.5% moderate, 10.3% severe
23	(Al-Shannaq et al., 2021) [47]	(FCV–19S) (DASS-21)	(41.4% with high level of fear towards COVID-19), (41.8%, 24.5% and 22.8% with mild, extreme and severe respectively on the depression, anxiety and stress symptoms)
24	(Al-Shannaq et al., 2021) [57]	(FCV–19S) (DASS-21)	N/A
25	(Al-Ajlouni et al., 2022) [41]	(IPAQ) (GAD-7) (CES-D) (PSQI)	(Prevalence of anxiety symptoms; 33.8% mild, 12.9% moderate, 6.3% severe), (Prevalence of depression symptoms: 21.5% highest quartile, 26.8% third quartile, 24.8% second quartile, 26.9% lowest quartile)
26	(Khatatbeh et al., 2021) [42]	A semi-structured individual interview to collect data. In-depth interviews were conducted with participants and recorded. Each interview lasted between 45–60 min.	Increased stress level, increased feeling of fear and anxiety, overall psychological stress
27	(Abu Kamel et al., 2021) [31]	(VAS) (IES-R) (FCV-19S) (PHQ-9)	Overall, the study indicated that COVID had moderate impact on symptoms of PTSD, Older adults with comorbidities has higher level of PTSD symptoms, 77.8% of PTSD as measured by IESR was explained by both FCV-19 and Depression (PHQ-9)
28	(Massad at al., 2020) [51]	(BAI)	Prevalence of anxiety: mild (21.5%), moderate: (10.9%), severe (6%)
29	(Akour et al., 2020) [52]	(K10)	Severe distress: 31.4%, Moderate: 17.5%, Mild: 20.7%, No distress: 30.4%
30	(Hawari et al., 2021) [53]	(K-6) (PROMIS)	Severe distress: 20%, high: 32%, Moderate: 17.5%, Mild: 20.7%, No distress: 30.4%
31	(Abuhamdah et al., 2021) [54]	(SRQ-20) (SSQ)	Prevalence of symptomatic mental distress was 65.7%

## 4. Discussion

### 4.1. Overview

This review assessed the impact of the COVID-19 pandemic on the mental health of the general population in Jordan. It also explored the related risk and predictive factors. In general, it is well established now that the COVID-19 pandemic had an impact on all areas of health. It is one of the biggest crises that affected the health system globally [58]. Mental health is one of the areas that has been enormously influenced due to the measures that have been taken to combat the global spread of the virus and all the economic consequences thereafter.

The results showed an increased prevalence of PTSD, psychological disorders, symptoms of anxiety, stress and depression among the different population groups in Jordan. Most studies showed an increase in the prevalence of symptoms related to mental health disorders; results varied from a moderate to a severe impact. The effect was not specific for any population group; it was universal among the general population. These results conformed with the global results on the impact of the COVID-19 pandemic on mental health. A study by Yunitri et al. [59] concluded that the COVID-19 pandemic caused a measurable impact on PTSD and mental health-related disorders globally and on the different population subgroups. The results from Jordan also conformed with other lower-middle-income countries (LMICs). In a CS research from Bangladesh by Das et al. [60] (p. 1), the study highlighted that ‘the prevalence of loneliness, depression, anxiety and sleep disturbance was estimated at 71%’. However, the results from Jordan were unique and represented strong data since Jordan had one of the most stringent lockdowns worldwide and a nearly zero caseload at the beginning of the pandemic [61].

Overall, the sample size of all studies was acceptable but not all were justified. This was due to the difficulty in collecting surveys during the pandemic period. Most of the surveys conducted were online. During the pandemic this method of sampling was convenient and reliable. However, this may have created some bias in the results of prevalence. Since the recruitment of participants was not stratified, results may not represent the real prevalence rates among the population holistically.

Jordan hosts a good number of refugees and displaced communities; two studies addressed the prevalence of mental disorders in those settings. First, the RCT conducted a pre- and post-pandemic assessment; it showed that pre-existing mental issues might not cause worsened psychological distress post-pandemic. The results were unforeseen yet justifiable as (1) refugees and displaced people are already suffering from forms of PTSD and the effect of this suffering is substantial [62]; (2) Syrian refugees in Jordan live in one of the second largest refugee camps worldwide, and during the strict lockdown period, mobility in and out of the camp was halted to minimise the risk of infections in such a densely populated area. However, these data and results are not enough to generalise.

The HCWs subgroup assessment showed a high prevalence rate, specifically at the beginning of the pandemic and when dealing with patients infected with COVID-19. This risk of high prevalence did not change even after one year of the pandemic. It is clear that HCWs were one of the most affected population subgroups. Being on the frontline of combating the pandemic, shortage of staff, full hospital capacity, and burnouts were all examples of how the pandemic affected the overall health services and system. The results from Jordan were similar to results from other countries. A study by Fournier et al. [63] included 77 hospitals in France and showed that the pandemic made devastating effects on HCWs. Even in a low-caseload country like Cyprus, HCWs reported a high level of mental health issues [64].

From the 31 studies reviewed, 3 reported results from elderly subgroups, with a total sample size of 1019. This confirmed that elderly people were underrepresented. None of the studies investigated the frail elderly group.

### 4.2. Risk Factors Associated with Higher Prevalence

#### 4.2.1. Gender Predictors

There was a consistency in results of being female as a predictor for having higher prevalence rates of mental health-related symptoms. Most studies concluded that women were experiencing more mental health problems than men in Jordan. Global research also confirmed that being female has a higher incidence of mental health-related issues. The higher incidence of mental disorders is gender-related and has been confirmed in studies even before the pandemic [65]. This may be due to different socio-economic, genetic and health-related factors. Males tend to have more substance abuse or antisocial disorders.

In LMICs and the refugee setting, this gender predictor can be more evident since females are more exposed to gender-based violence and sexual assault, are paid less for work and get divorced, and other factors. It is well documented that the female population requires health-specific needs apart from general needs. Although the research in this area is still limited, it showed that the effect of the pandemic on females was higher on the mental health front [66].

#### 4.2.2. Socio-Economic Predictors

The pandemic imposed economic and social hardship on the population in general. Jordan was already suffering from that hardship pre-pandemic: unemployment, loss of work and youth unable to find work, in addition to limited financial support [67]. On the social level, lockdowns caused physical distancing, more use and addiction to social media, less human interactions and a lack of proper social support. All these socio-economic factors are considered to be predictors for mental health disorders. Most of the reviewed studies indicated a correlation between the increased prevalence of mental health issues and socio-economic situation, as per Gong et al. [68] (p. 11)

*‘Loneliness, insecurity, anxiety, depression, sleep problems, discrimination, and substance abuse are adverse mental consequences experienced by individuals experiencing economic turmoil during the pandemic’*.

This predictor is key since the effect of the socio-economic hardship is still ongoing globally and in Jordan. It is evident that more population groups are in need of financial and social support to be able to overcome the pandemic impact. However, this requires more research and looking into the long-term effects of the pandemic on the affected population groups.

#### 4.2.3. Comorbidities Predictors

Data showed that people suffering from chronic diseases or pre-psychiatric conditions have higher levels of prevalence of mental health symptoms. The combination of clinical or physical illness with a mental disorder is considered a risk factor, and this is an area of big concern [69]. It worsens if comorbidities are affecting the elderly, refugees or vulnerable community members. These results corresponded with other global studies that confirmed that the prevalence of anxiety and depression symptoms and the level of stress were significantly higher among adults with comorbidities [70].

### 4.3. Recommendations and Lessons Learned from the Impact on Mental Health

The impact of the COVID-19 pandemic was an opportunity for elevating and enhancing mental health services. The pandemic had put mental health at the forefront of the public health crises. This may have not been the case if the situation was still pre-pandemic. In Jordan there have been focused efforts in the last period post-pandemic to enhance health services in general and mental health in specific; it is recommended that mental health services be more integrated within primary health services [71].

The pandemic has also been an opportunity to further the research in the field of mental health, which will surely reflect upon the entire research of mental health in the country, but it is also recommended to further advance this research in areas such as refugee settings, elderly population and how to strengthen mental health services within the health system. These focus areas are discussed in the national mental health and substance abuse plan of Jordan [72].

HCWs were one of the most affected groups as the lack of proper mental health training programs and psychiatric educational courses were among the factors that influenced this group, in addition to the shortage of human resources and HCWs specialised in mental health and psychiatric illnesses [73]. It is recommended to increase and advance educational and training services for mental health providers in Jordan.

On the community level, taking the case of university students as an example, this review showed a lack of awareness and educational programs in universities, which has led students to be more likely affected by the impact of COVID-19 on their mental health. The disturbance that occurred due to the dramatic shift to virtual and online education also caused an increasing impact on the mental well-being of students and teachers alike. It is recommended to expand community awareness programs to support the onset of mental health disorders.

On the refugee front, it is recommended to incorporate psychosocial education in schools and universities for elevating awareness and building resilience. Such programs are of great impact on enhancing the resilience of refugees to better incorporate within their communities and cope with mental health issues [74]. Piloting the idea and re-searching it in vulnerable communities would be of great importance.

Finally, on the country policy level, more governance and implementation of mental health and psychological support programs are needed to make sure that the health system is resilient and adaptable to any future crisis.

### 4.4. Strengths and Limitations

#### 4.4.1. Strengths

To the best of our knowledge, this is considered the first systematic review conducted for Jordan that addressed the impact of COVID-19 on the mental health of the general population. Moreover, this review addressed the outcomes and predicting factors, while also suggesting recommendations on how to tackle mental health issues at the country level.

#### 4.4.2. Limitations

This review was conducted on a descriptive narrative analysis and not a meta-analysis. This for sure affects the overall assurance of results and the measurement of the real impacts from the reviewed publications. Also, the variances in study subgroups, more representation of some groups such as students, younger age, females and less representation of others such as elderly, adolescents and refugees, have imposed a limitation on generalising the outcomes. Finally, including all studies regardless of quality appraisal outcomes may have created some bias as it did not address comparable studies.

## 5. Conclusions

This systematic review assessed the impact of the COVID-19 pandemic on the mental health of the population in Jordan. The results showed a significant correlation between the pandemic and the effect it imposed on mental health. Increased levels of PTSD/ psychological distress/ anxiety, depression and stress were reported among the different population groups. The lessons learned from the impact have shed light on several recommendations on how to address this mental health epidemic to help Jordan overcome the effect and be able to build a more robust strategy for the future.

## Figures and Tables

**Figure 1 ijerph-20-06382-f001:**
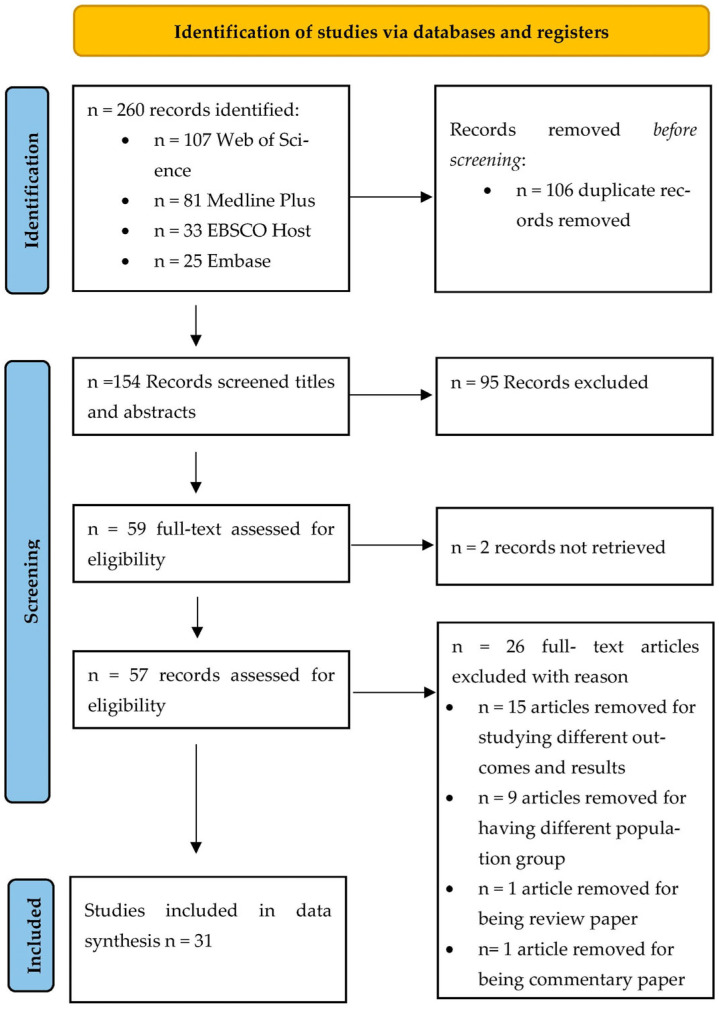
Preferred Reporting Items for Systematic Reviews and Meta-Analysis (PRISMA) study selection flow diagram [18].

**Figure 2 ijerph-20-06382-f002:**
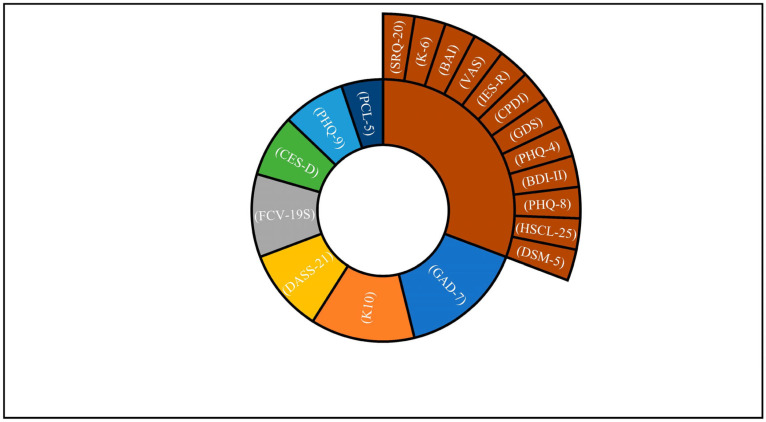
Assessment tools of psychological distress/ PTSD/ anxiety, depression and stress.

## Data Availability

No new data were created or analyzed in this study. Data sharing is not applicable to this article.

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
