# Peer review of "Impact of COVID-19 Pandemic on Mental Health among the Population in Jordan"

_ijerph, 2023, doi:10.3390/ijerph20146382_

Round 1

Reviewer 1 Report

Thank you for the opportunity to review this systematic review of mental health in Jordan. It is impressive that this many articles were identified. The methodology is straightforward and tables/figures clearly presented.

Improvement is needed in the following ways:

Abstract: conclusion that the pandemic had a significant impact on mental health is not substantiated by the results provided in the abstract.

Introduction: references are missing to provide evidence for the statements being made, for example: 

The COVID-19 pandemic hit the world by 2021, making it the worst health crisis that humans have faced in recent history, especially in terms of mental health (references missing here).

Worldwide, various studies have reviewed the impact of COVID-19 pandemic on the population level (provide references here, and several have studied the impact on sub-population groups such as 50 Healthcare Workers (HCW’s) (references here) and refugees (references here) and summarise these.

Jordan took stringent measures to combat the pandemic. What were these, provide more information so the reader understands the context.

Methods:

- has the PRISMA checklist been provided as Appendix?

Discussion:

- you state in the introduction that a further objective was to give recommendations and you state this as a strength. Where are these recommendations as section 2.3 is only lessons learnt. In this section it is again very general statements without references to provide evidence of what is being stated, e.g. 'further the research in mental health' how is this being done?

Conclusion: where is the evidence that the case of Jordan is similar to that of LMICs, as this was not the objective of the current study and is outside the scope to state this.

Minor errors throughout that are easy to correct, e.g. page 1 line 19 abstract lower case p for pandemic, line 2 health care workers first letter of each word not upper case, line 52 same page 'there was (past tense) a shortage, line 59 same page, sentence should start with 'A high prevalence rate of stress....

page 17 line 410: On the community level, and in the case of university students. This / remove period and place comma, t should be lower case.

Author Response

Thanks for the review and your comments. Please see the attachment for the reply on your comments raised. 

Reviewer 2 Report

The introduction is well structured and addresses contextualization, state of the art, gap and justifications of the study. The description of the methods is clear and replicable. Also responds to the objective of identifying the impact of the COVID-19 pandemic on the mental health of the Jordanian population. Methods and results were formatted based on PRISMA reporting guidelines. Tables are very appropriate, as well as Figure 2 is quite illustrative about the most used mental health assessment instruments in Jordanian. The discussion is pertinent and based on the 31 papers included in the review. I recommend approval as it is written. Congratulations to the authors.

Author Response

Thanks for your review. Please see attachment.  

Reviewer 3 Report

This is a study that clarified mental health and its factors during a pandemic in Jordan through a literature review. It is commendable that the authors spent an enormous amount of time organizing many papers. However, the conclusions drawn are quite general. Although he mentions the large number of refugees as a characteristic of Jordan, there are only a few papers that deal directly with refugees. Furthermore, the finding that women, students, sick people, and people with low social status suffered greater mental damage from infection was not new, consistent with reviews from other countries. These results lower the rating of this paper. To further enhance the value of this paper, the authors should emphasize the originality of the review. To that end, the authors should enrich the discussion on what kind of psychological damage the refugees suffered. Authors may add non-article material, such as news media, to these discussions. In addition, it should point out what is lacking in the existing papers and indicate the direction in which future research should aim.

Author Response

Thanks for your review. Please see attachment for reply on your comments. 

Round 2

Reviewer 1 Report

Comments adequately addressed.

Minor language editing needed.

Reviewer 3 Report

Enough improvements have been made. I agree to the posting.